# A Hydrolyzed Chicken Extract CMI-168 Enhances Learning and Memory in Middle-Aged Mice

**DOI:** 10.3390/nu11010027

**Published:** 2018-12-22

**Authors:** Sheng-Feng Tsai, Chia-Yuan Chang, Shan-May Yong, Ai-Lin Lim, Yoshihiro Nakao, Shean-Jen Chen, Yu-Min Kuo

**Affiliations:** 1Institute of Basic Medical Sciences, College of Medicine, National Cheng Kung University, Tainan 70101, Taiwan; eric04142000@hotmail.com; 2College of Photonics, National Chiao Tung University, Hsinchu 30010, Taiwan; sheanjen@nctu.edu.tw; 3Advanced Optoelectronic Technology Center, National Cheng Kung University, Tainan 70101, Taiwan; cychang0829@gmail.com; 4Scientific Research and Applications, BRAND’S Suntory, Singapore 048423, Singapore; ShanMay.Yong@Brands-Suntory.com (S.-M.Y.); Aileen.Lim@Brands-Suntory.com (A.-L.L.); Yoshihiro.Nakao@Brands-Suntory.com (Y.N.); 5Department of Cell Biology and Anatomy, College of Medicine, National Cheng Kung University, Tainan 70101, Taiwan

**Keywords:** essence of chicken, hippocampus, neuroadaptation, neuroplasticity, anti-stress

## Abstract

There has been increasing evidence that consumption of dietary supplements or specific nutrients can influence cognitive processes and emotions. A proprietary chicken meat extraction, Chicken Meat Ingredient-168 (CMI-168), has previously been shown to enhance cognitive function in humans. However, the mechanism underlying the CMI-168-induced benefits remains unclear. In this study, we investigated the effects of CMI-168 on hippocampal neuroplasticity and memory function in middle-aged (9–12 months old) mice. The mice in the test group (termed the “CMI-168 group”) were fed dietary pellets produced by mixing CMI-168 and normal laboratory mouse chow to provide a daily CMI-168 dose of 150 mg/kg of body weight for 6 weeks. The control mice (termed the “Chow group”) were fed normal laboratory mouse chow pellets. CMI-168 supplementation did not affect the body weight gain, food intake, or exploratory behavior of the mice. In the novel object recognition test, the CMI-168 group showed better hippocampus-related non-spatial memory compared to the control Chow group. However, spatial memory examined by the Morris Water Maze test was similar between the two groups. There was also no significant difference in the induction and maintenance of long-term potentiation and dendritic complexity of the hippocampal cornu ammonis region 1 (CA1) neurons, as well as the levels of neuroplasticity-related proteins in the hippocampi of the CMI-168 and Chow groups. Interestingly, we observed that CMI-168 appeared to protect the mice against stress-induced weight loss. In conclusion, dietary supplementation of CMI-168 was found to improve learning and memory in middle-aged mice, independent of structural or functional changes in the hippocampus. The resilience to stress afforded by CMI-168 warrants further investigation.

## 1. Introduction

Learning and memory involves a series of processes in which external information is encoded, stored, and retrieved. It is widely assumed that the encoding and storing of memory traces in the central nervous system is accompanied by changes in the strength of neuronal connections, also known as neuroplasticity [1,2]. Neuroplasticity is induced at appropriate synapses during memory processes and is necessary for information storage. More specifically, in declarative (or explicit) memory, the hippocampus plays an essential role [3], and structural or functional changes in this brain region are related to learning and memory performance [4].

In a previous placebo-controlled, double-blind randomized clinical study, Azhar and colleagues showed that the proprietary chicken meat ingredient CMI-168 had cognition-enhancing benefits [5]. In the study, healthy subjects aged 35 to 65 years given CMI-168 for six weeks demonstrated enhanced performance in cognitive tests that assessed the subjects’ learning and memory. This effect was shown to be sustained even two weeks after the termination of CMI-168 supplementation, although the underlying mechanism remains unclear.

Herein, we investigate the effects of CMI-168 on the hippocampus-related neuroplasticity and memory functions in middle-aged mice at the behavioral, cellular, and molecular levels to provide mechanistic insights of CMI-168 in memory. Middle-aged (9–12 months old) mice were used in this study because this is the age in mice when memory impairment starts to show. After six weeks of CMI-168 supplements, mice were tested for changes in hippocampus-related spatial memory using the Morris water maze (MWM) test [6] and for hippocampus-related non-spatial memory using the novel object recognition (ORT) test [7]. Hippocampal neuroplasticity was determined by using long-term potentiation (LTP) induction and maintenance, which is a subcellular model for synaptic events underlying memory formation. Additionally, we examined for any changes to hippocampal CA1 neuron morphology or neuroplasticity-related protein levels in the hippocampus.

## 2. Materials and Methods

### 2.1. CMI-168

The Chicken Meat Ingredient (CMI-168) used in this study was a hydrolyzed chicken extract prepared from chicken meat that had undergone proprietary bio-processing technology and aqueous extraction. CMI-168 consists of the following basic nutrients: 91.4% protein (4.2% free amino acids, 7.6% diketopiperazines or cyclic dipeptides), 1.0% carbohydrate, 1.6% lipid, 2.0% moisture, and 4.0% minerals and ash.

### 2.2. Animals

All experiments were done in accordance with the National Institute of Health Guideline for Animal Research (Guide for the Care and Use of Laboratory Animals) and approved by the National Cheng Kung University Institutional Animal Care and Use Committee (approval number: 105269). Male C57BL/6 mice, 9–12 months old, were purchased from National Laboratory Animal Center (Nangang, Taipei, Taiwan) and randomly assigned to the normal laboratory mouse chow (Chow) or Chow + CMI-168 (CMI-168)-supplemented groups. Mice were housed 4–5 per cage under a 12 h light/12 h dark cycle (lights on at 8:00 a.m.) and at constant temperature (24 ± 1 °C) and humidity in a controlled room at the National Cheng Kung University Animal Center. Food and water were made available *ad libitum* to the mice.

Based on the efficacious dose of 679 mg/kg body weight/day used in a previous human study [5], this was converted to a mouse-equivalent dose of 150 mg/kg body weight/day by using a conversion factor of 12.3 [8]. The CMI-168 pellets were prepared by mixing both CMI-168 powder and powder ground from normal laboratory mouse chow pellets (5010, LabDiet, St. Louis, MO, USA). The mixture was reconstructed into pellets to be included in the mouse diet. The ratio of the amount of CMI-168 powder to the mouse chow powder was based on the average daily food intake of the mice calculated one week before the beginning of the experiment. Mice from the CMI-168 group were fed with the reconstructed pellets containing CMI-168 for six weeks, while the Chow mice were fed normal reconstructed pellets for the same period. One day after the six-week feeding period, mice were divided into three separate batches for different studies and analyses: Batch 1 was designed to examine the learning and memory performance, including that of the novel object recognition test (ORT), Morris Water Maze (MWM), and reversal MWM; Batch 2 was designed to examine the hippocampal expression of neuroplasticity-related proteins; and Batch 3 was used to examine neuron morphology and LTP induction. The CMI-168 mice were kept on a CMI-168 diet throughout the behavioral tests. The daily food consumption and weekly body weight of these mice were recorded.

### 2.3. Open Field Test

The protocol of the open field test (OFT) was modified from a previous study [9]. The spontaneous activity in a novel environment was quantified by introducing the mice into a polycarbonate box (50 × 40 × 40 cm) for 15 min. The time spent in the central zone (25% of the surface area) and the number of entries into the central zone were measured and analyzed.

### 2.4. Novel Object Recognition (ORT) Test

The ORT was used to determine the hippocampus-related non-spatial learning and memory, as previously described [10]. After being habituated to a polycarbonate box (50 × 40 × 40 cm) for 10 min per day over three consecutive days, the mice were returned to the same box containing two identical objects (250 mL glass beakers, 6.5 cm in diameter and 9 cm in height, placed upside down) each separately positioned 5 cm away from a wall. The cumulative time spent by the mouse in exploring each of the objects was recorded manually over a 5 min period. The baseline trial performance was indicated as the ratio of the time spent exploring one of two identical objects over the total exploring time. Two hours later, the mouse was reintroduced into the box for the short-term memory (STM) test. One of the two objects was randomly replaced by a new one (small glass bottle, 2.8 cm in diameter and 6.5 cm in height). For the long-term memory (LTM) test, the mouse was reintroduced into the box after 24 h. One of the two objects was replaced by a new one (small iron bottle, 3.5 cm in diameter and 7.2 cm in height). The time spent exploring each object over a 5 min period was recorded for both the STM and LTM tests. All of the objects were cleaned with 70% alcohol between trials to avoid olfactory cues. The exploratory behavior was defined as “mice touching the object with the nose or sniffing toward the object within a distance of 1 cm”. We used the ratio of “new object exploring time divided by total exploring time” to represent the memory function.

### 2.5. Morris Water Maze (MWM) Test

The MWM test was performed using a circular pool with a diameter of 110 cm and a wall height of 60 cm, as previously described [11]. The pool was filled to a depth of 35 cm with clear, 24 ± 1 °C tap water. The circular escape platform made of transparent Plexiglas (diameter: 10 cm) was submerged 1 cm below the surface of the water. During all the trials involving spatial navigation, the location of the hidden platform was kept constant. The mice were given one-session training beginning at 10:00 a.m. per day for five days. Each session consisted of four swim trials (maximum 120 s per trial) with different quadrant starting positions for each trial. 24 h after the last training session, the mice were placed in the southwest position of the pool—the longest distance from the platform position—and were allowed to swim for 60 s without a platform present.

For the reversal MWM test, the reversal learning test was performed by relocating the platform in a different quadrant and repeating the procedures of spatial navigation and the probe test. The whole process was recorded by a charge-coupled device camera, and the escape latency (i.e., time taken to reach the platform, in seconds) and time spent within the target quadrant in the probe test were analyzed using a video tracking system (EthoVision; Noldus Information Technology, Wageningen, Netherlands).

### 2.6. Electrophysiology

The brains of mice were quickly removed, placed in chilled artificial cerebrospinal fluid (aCSF, 117 mM NaCl, 4.7 mM KCl, 2.5 mM CaCl_2_, 11 mM glucose, 1.2 mM MgCl_2_, 25 mM NaHCO_3_, 1.2 mM NaH_2_PO_4_, pH 7.4) (Tocris Bioscience, Bristol, UK) and equilibrated with 95% O_2_ and 5% CO_2_. Using a vibratome (DTK-1000N, Dosaka, Kyoto, Japan), the brains were cut into 400 μm transverse slices and recovered for 1 h at room temperature in aCSF bubbled with a mixture of 95% O_2_ and 5% CO_2_. A 64-channel microelectrode array system (MED-P515A, Alpha MED Scientific Inc., Osaka, Japan) was used to record the field excitatory postsynaptic potentials (fEPSP) and introduce high-frequency stimulation. The brain slices were placed into a submersion chamber equipped with the 64-channel multielectrode probe with the CA1 pyramidal cell layer positioned directly over the array contacts. Recordings were made at 37 °C in circulating aCSF containing 10 μM bicuculline and bubbled with 95% O_2_ and 5% CO_2_. The fEPSP were captured and analyzed using a packed software v1.4.5 (Mobius software, Alpha MED Scientific Inc., Ibaraki, Osaka, Japan). There was a 10 min period of pre-induction baseline measures in which stimuli were elicited at 20 s intervals. After the baseline measurements, LTP was induced by one train of high-frequency stimulation (100 Hz, 1 s). The fEPSPs were recorded for at least 60 min after the high-frequency stimulation had begun. Prior to the start of the experiment, stimulation intensity was determined as the intensity producing a fEPSP amplitude which was 60% of the maximum response. Each piece of data was obtained from one single hippocampal slice from different mice.

### 2.7. Single-Neuron Labeling

One day after the termination of CMI-168 supplementation, one set of mice were prepared for the neuronal morphology assay by single-neuron labeling, as described previously [12]. The anesthetized mice were perfused from the left ventricle with 50 mL of aCSF followed by 4% buffered paraformaldehyde (30 mL/40 g body weight). The brains were dissected, post-fixed with 4% buffered paraformaldehyde at 4 °C for 3 h, and then coronally sliced into 400 µm sections using a vibratome (DTK-1000N, Dosaka, Shizuichi, Kyoto, Japan). The sections were viewed under differential interference contrast optics to identify neuronal somas in the CA1 area. Neuronal somas were impaled with an intracellular electrode filled with saturated lucifer yellow solution which was delivered iontophoretically to fill the entire neuron. The labeled slice was then dehydrated in saturated sucrose solution and mounted with the medium (80% glycerol and 20% 20 mM sodium carbonate). A fluorescence microscope was used to identify and trace the labeled dendrites. The dendritic field of the fluorescent dye-labeled neuron was measured using ImageJ 1.51v (National Institutes of Health, Bethesda, MD, USA) software’s plugin, NeuronJ v1.4.3 (National Institutes of Health, Bethesda, MD, USA), and Sholl analysis v3.7.3 (National Institutes of Health, Bethesda, MD, USA). Five neurons in five sections from each mouse brain were labeled and measured.

### 2.8. Immunoblotting

The relative expression of brain-derived neurotrophic factor (BDNF), full-length/truncated tyrosine receptor kinase kinase B (FL-TrkB/T-TrkB), synaptotagmin-1 (SYT-1), SYT-4, synaptosomal-associated protein-25 (SNAP-25), Ca^2+^/calmodulin-dependent protein kinase II (CamKII), extracellular regulated protein kinase 1/2 (Erk1/2), protein kinase C-ζ (PKC_ζ_), glutamate receptor 1 (GluR1), p38 mitogen-activated protein kinase (p38), glycogen synthase kinase 3-α/β (GSK3α/β), and the phosphorylated levels of CamKII, Erk1/2, p38, and GSK3α/β in the hippocampus were determined using immunoblotting.

The anesthetized mice were perfused from the left ventricle with chilled phosphate-buffered saline, and their brains were quickly removed. The hippocampus specimens were dissected and homogenized in ice-cold commercial tissue protein extraction reagent (78510, Thermo Fisher Scientific Inc., Waltham, MA, USA) containing protease and phosphatase inhibitors (04693116001 & PHOSS-RO, Roche Diagnostics, Mannheim, Germany). The homogenates were centrifuged at 13,000× *g* for 15 min at 4 °C, and the protein concentrations of the supernatants were determined and adjusted to the same concentration. Supernatants (10 μg of total protein) were mixed with sample buffer that contained 2% of 2-mercaptoethanol, loaded into each well of 8–15% polyacrylamide gel and resolved at 110 V for 2 h. The separated proteins were transferred to a polyvinylidene fluoride membrane (IPVH00010, MerckMillipore/Merck KGaA, Darmstadt, Germany), blocked with 5% milk and hybridized with primary antibodies overnight at 4 °C. The information of primary antibodies is described in Table 1. After washing, the membranes were hybridized with proper horseradish peroxidase-conjugated secondary antibodies (Jackson ImmunoResearch Inc., West Grove, PA, USA). The bound antibodies were detected using an enhanced chemiluminescence detection kit (WBKLS0500, MerckMillipore/Merck KGaA, Darmstadt, Germany). Relative protein expression was estimated by normalizing with the β-actin levels. The band densities were measured using an imaging system (BioChemi; UVP, Upland, CA, USA) and analyzed using ImageJ 1.51v (National Institutes of Health, Bethesda, MD, USA).

### 2.9. Statistical Analysis

Data are expressed as mean ± standard error of the mean (SEM). Significance was set at *p* <0.05. The gains in body weight, food consumptions, and escape latencies in the learning sessions of the Morris water maze were analyzed by repeated measured two-way ANOVA with time (ages or experimental time points) and CMI-168 supplementation as the main factors. The unpaired, two-tailed Student’s *t*-test was used for the rest of the analyses with CMI-168 as the single independent variable factor. The Sholl analysis were analyzed by two-way repeated measures (mixed-model) ANOVA with concentric rings (increasing diameters of 10 µm) and CMI-168 supplementation as the two main factors. Bonferroni’s *post-hoc* test was used to perform multiple comparison analysis after the two-way ANOVAs. Detailed sample sizes for each experiment are labeled in the figures.

## 3. Results

### 3.1. CMI-168 Did Not Affect Body Weight or Food Consumption in Middle-Aged Mice

The weekly body weight and daily food consumption during the six-week feeding period were monitored. Results showed that the body weights of the Chow and CMI-168 mice were unchanged during the feeding period [*F* = 0.95, degree of freedom (*d.f.*) 6/336, *p* = 0.46, time factor, repeated two-way ANOVA, Figure 1A]. However, the food intakes were found to have altered during the feeding period (*F* = 2.60, *d.f.* 5/50, *p* = 0.0363, time factor, repeated two-way ANOVA, Figure 1B).

### 3.2. CMI-168 Enhanced the Hippocampus-Related Non-Spatial Memory in Middle-Aged Mice

ORT was used to examine the effects of CMI-168 on hippocampus-related non-spatial memory. As anxiety is able to suppress the desire to explore [13], we first used the open field test to investigate any anxiety-like behavior of the mice. The results showed that the CMI-168 and Chow mice spent comparable time in the centre zone (*p* = 0.52, Student’s *t*-test) (Figure 2A). The numbers of entries into the centre zone were also no different between the two groups (*p* = 0.95, Student’s *t*-test).

In the acquisition phase of ORT, the percentage of time spent exploring either of the two objects was around 50% for both the CMI-168 and Chow groups (*p* = 0.45, Student’s *t*-test) (Figure 2B, Baseline). This suggested that the mice had no place preference for either of the two objects. Six weeks of CMI-168 supplementation was able to enhance both the STM (*p* = 0.0006, Student’s *t*-test) and LTM (*p* = 0.0355, Student’s *t*-test) (Figure 2B) of the mice, as measured by the increase in % time spent exploring the novel object.

### 3.3. CMI-168 Did Not Alter the Hippocampus-Related Spatial Memory in Middle-Aged Mice

The MWM test was used to examine the effects of CMI-168 on hippocampus-related spatial memory. Results showed that the escape latencies of both Chow and CMI-168 mice reduced, session by session (*F* = 79.98, *d.f.* 4/144, *p* < 0.0001, session factor, repeated two-way ANOVA, Figure 3A). The *post-hoc* analysis revealed that the escape latencies in the fifth session were significantly less than those in the first session in both the Chow (*p* < 0.0001) and CMI-168 (*p* < 0.0001) groups. These data indicated that both CMI-168 and Chow mice successfully learned the location on the platform in the MWM test (Figure 3A). After six weeks of feeding, the escape latencies in the learning sessions were similar between the CMI-168 and Chow groups (*F* = 0.04, *d.f.* 1/36, *p* = 0.84, repeated two-way ANOVAs, Figure 3A). In the probe test, the time spent in the platform quadrant was also similar between two groups (*p* = 0.80, Student’s *t*-test, Figure 3A). In the reversal MWM test, which was adopted to test the behavioral flexibility, both Chow and CMI-168 mice also successfully learned the platform location by showing decreased escape latencies session by session (*F* = 13.38, *d.f.* 4/72, *p* < 0.0001, session factor, repeated two-way ANOVA, Figure 3B). *Post-hoc* analysis revealed that the escape latencies in the fifth session were significantly less than those in the first session in both Chow (*p* = 0.0002) and CMI-168 (*p* = 0.0001) groups in the reversal MWM test (Figure 3B). Moreover, CMI-168 and Chow mice showed comparable escape latencies in the learning sessions (*F* = 0.02, *d.f.* 1/18, *p* = 0.89, repeated two-way ANOVA) and time in the platform quadrant in the probe test (*p* = 0.80, Student’s *t*-test) (Figure 3B).

Interestingly, we found that the body weights of the Chow mice decreased significantly after the six days of MWM tests (five days in learning sessions and one day in probe test), whereas the body weights of the CMI-168 mice were unaffected by the test (*p* = 0.0002, Student’s *t*-test, Figure 3C). These results suggested that CMI-168 may protect middle-aged mice against swimming stress-induced weight loss.

### 3.4. CMI-168 Did Not Alter the LTP Induction, LTP Maintenance or the Dendritic Complexity of the Hippocampal CA1 Neurons in Middle-Aged Mice

We measured long-term potentiation (LTP), a major cellular mechanism that underlies learning and memory [14], to investigate the synaptic transmissions in the Schaffer collateral-CA1 synapse of the hippocampus. High-frequency stimulation successfully induced LTP in middle-aged mice of both the Chow and CMI-168 groups (*F* = 11.16, *d.f.* 35/245, *p* < 0.0001, time factor, repeated two-way ANOVA, Figure 4). The fEPSP slopes were similar between these two groups (*F* = 0.01, *d.f.* 1/7, *p* = 0.93, repeated two-way ANOVA, Figure 4).

The effects of CMI-168 on hippocampal CA1 neuron morphology was also examined using the single-cell labelling technique by injecting fluorescent dye (lucifer yellow) into the somas of individual neurons. The labeled dendrites were traced (Figure 5A) and subjected to Sholl analysis. The results showed that the numbers of intersections between dendrites and concentric rings were comparable between CMI-168 and Chow groups (*F* = 1.86, *d.f.* 1/128, *p* = 0.17, mixed model two-way ANOVA, Figure 5B). These results suggested that CMI-168 did not affect the transmission function or structure of the hippocampal CA1 neurons.

### 3.5. CMI-168 Did Not Alter the Hippocampal Expression of Neuroplasticity-Related Molecules in Middle-Aged Mice

The effects of CMI-168 on the expression and phosphorylation status of selected neuroplasticity-related proteins in the hippocampi of the middle-aged mice were determined by Western blot (Figure 6). There were no differences in the relative expression levels of BDNF, FL-TrkB, T-TrkB, SYT-I, SYT-IV, SNAP-25, PKC_ζ_, and GluR1 between the CMI-168 and Chow groups (Table 2****). The relative levels of phosphorylated CamKII, Erk1, Erk2, p38, GSK3α(Y^279^), GSK3α(S^21^), GSK3β(Y^216^), and GSK3β(S^9^) were also comparable between these two groups (Table 2).

## 4. Discussion

It has been reported that the essence of chicken enhances the maintenance of our brain’s executive functions, including attention, short-term and working memory, and especially under stressful situations in humans [15]. To further explore the bioactive ingredients responsible for the observed benefits in cognitive function, a chicken meat ingredient, CMI-168 was developed and shown to improve the working memory, short-term, and verbal learning memory in healthy middle-aged human subjects [5]. Improved performance in the cognitive tests was seen within six weeks of CMI-168 supplementation, and maintained even two weeks after termination of supplementation. We adopted the parameters of CMI-168 supplementation based on these findings, including dose, duration, and subject age for use in this study to explore the mechanisms underlying the benefits of CMI-168 on the cognitive functions in mice.

The hippocampus, a cognition-related region, was chosen as the assay target due to its relatively clear anatomical (e.g., neural transmission circuitry) and physiological (e.g., governing spatial navigation and spatial-related memory formation) properties. Results showed that CMI-168 improved the performance of the mice in the ORT, both in regard to short-term and long-term memory, corresponding with the improvements in learning and memory seen in the clinical study [5]. In line with this present study, we recently treated senescence-accelerated mice with long-term CMI-168 supplementation at dosages of 150, 300, and 600 mg/kg, and found that all dosages enhanced learning and memory in passive avoidance and active shuttle response tasks without obvious adverse side-effects [16]. However, we did not find differences in the hippocampal neuroplasticity-associated parameters, including LTP induction and maintenance in the hippocampal Schaffer collateral-CA1 pathway, dendritic complexity of the hippocampal CA1 neurons, or changes in expression or phosphorylation levels of hippocampal neuroplasticity-related proteins between the CMI-168 and Chow mice. Among the selected neuroplasticity-related proteins, BDNF-TrkB signaling [17,18], CamKII [19], Erk1/2 [20], PKCζ [21], GluR1 [22], p38 [23], and GSK3 [24] have been linked to the learning and memory functions. SYT-1, SYT-4, and SNAP-25 are members of soluble *N*-ethylmaleimide-sensitive factor attachment protein receptor located in the presynaptic and postsynaptic regions. These synaptic proteins regulate the release of neurotransmitters at the presynaptic terminals and the localization of receptors in the postsynaptic terminals [25]. Reducing the local expression of SYT-1, SYT-4, and SNAP-25 is known to impair learning and memory [26]. Nonetheless, CMI-168 did not affect the expressions of these synaptic proteins, nor the neuronal structures in the hippocampus. Taken together, our results suggest that there may not be a direct involvement of the hippocampus in the enhancement of cognitive functions by CMI-168 supplementation.

Besides the hippocampus, the medial prefrontal cortex, medial dorsal thalamus, and perirhinal cortex are also critical areas for object recognition memory. This is in line with Azhar et al.’s findings [5] that CMI-168 supplementation promotes the cognitive functions related to the prefrontal cortex. In another study, the intake of chicken essence for seven days significantly increased oxy-hemoglobin levels in the dorsolateral prefrontal cortex, which was correlated with increased activity during working memory tasks [27]. Therefore, it may be necessary to characterize the neuroplasticity in other ORT-related brain regions in order to have a clear understanding of the action of CMI-168.

CMI-168 consists largely of proteins, peptides, and amino acids, including glutamic acid, arginine, tyrosine, and histidine. Among them, the amount of diketopiperazines or cyclic dipeptides are abundant (>7.5%) in the CMI-168. Diketopiperazines are natural compounds that have been detected endogenously in mammals [28,29] and in processed foods, such as chicken essence, roasted coffee, cocoa, and beer. These types of cyclic dipeptides can be formed through non-enzymatic dehydration and condensation of two N-terminal amino acid residues of linear peptides during food sterilization [30]. Some of the diketopiperazines and their derivatives have neuroprotective or nootropic effects [31,32]. For instance, endogenous cyclo-prolylglycine (cyclo-(Pro-Gly)) expressed in the rat brain, and cyclo(L-Phe-L-Phe) found in chicken essence exhibit anti-amnesic activity [29,30,33]. Cyclo-(Pro-Gly) prevents memory decline evoked by maximal electroshock (MES) [5,32], whereas cyclo(L-Phe-L-Phe) ameliorates scopolamine-induced learning and memory impairment in mice [30]. Tsuruoka et al. were able to purify and identify cyclo(L-Phe-L-Phe) in chicken essence [30]. They demonstrated that chronic oral administration of synthetic cyclo(L-Phe-L-Phe) could increase extracellular levels of the cerebral monoamines—serotonin, norepinephrine, and dopamine—in the medial prefrontal cortex, as well as acetylcholine in the ventral hippocampus of rats. In that study, cyclo(L-Phe-L-Phe) was able to inhibit both the acetylcholinesterase and the serotonin transporter, suggesting that this dual inhibitor function could be responsible for significantly improving depressed behavior in mice. Acetylcholine is a vasodilator of cerebral arteries and induces increase in cerebral blood flow velocity through a nitric oxide-dependent pathway [34,35]. Furthermore, patients with Alzheimer’s disease taking acetylcholinesterase inhibitors show better cognitive performance and increased regional blood flow in the frontal lobe [36]. Taken together, these findings suggest that CMI-168 may increase cerebral blood flow and brain activity in the prefrontal cortex, which in turn leads to an enhancement of learning and memory.

Besides cyclic dipeptides, several amino acids that are found in CMI-168 are known to play roles in facilitating learning and memory. Chronic oral administration of glutamate enhances recognition memory in rats via increasing hippocampal neurotransmitter levels, particularly acetylcholine, glutamate, and gamma-amino butyric acid (GABA) [37]. Another important amino acid found in CMI-168 is L-arginine (L-Arg), which is associated with many physiological processes, including cognitive, cardiovascular, sexual, and gastrointestinal tract functions [38]. It can be converted to NO and L-citruline by nitric oxide synthase, whereby the former can act as a non-noradrenergic and non-cholinergic transmitter to exert vasodilatation and neuroprotective effects [39,40,41]. It has been shown that L-arginine enhances cognitive function in elderly patients, possibly via reducing lipid peroxidation, which is an indicator of oxidative stress [42]. Other studies have also demonstrated the beneficial effects of L-arginine on learning and memory in various mouse models with learning deficits [43,44,45]. In addition, L-arginine is capable of reducing oxidative stress that contributes to cognitive decline during aging [46,47,48]. The regulation of oxidative stress, whether by decreasing markers of lipid peroxidation, such as thiobarbituric acid reactive substances, or by preventing depletion in the serum antioxidant enzyme level, is well-documented in clinical and animal studies [42,46,49,50]. This antioxidant function of L-arginine may be mediated via specific isoforms of NO, which inhibit the effects of inositol-1,2,5-triphosphates and lipoxygenase-dependent lipid and lipoprotein oxidation [50,51,52]. Other possible amino acids that may contribute to the effects of CMI-168 include tyrosine, a precursor of dopamine that improves working memory in human subjects [53,54,55], and ameliorated, hypoxia-induced learning and memory impairment in rats [56]. The effect of CMI-168 in facilitating learning and memory may be a result of specific amino acids or peptides, all potentially contributing to the efficacy of CMI-168. The exact mechanism and nature of the potential bioactive(s) would require further investigation.

Interestingly, although CMI-168 supplementation did not affect normal gains in body weight or feeding behaviors, it protected mice from repeated swimming stress-related weight loss. Repeated swimming stress is known to induce weight loss in both C57BL/6 and BALB/c mice [57]. In agreement with our findings, human clinical studies [58,59,60] and rodent studies [61,62] have also shown that chicken essence exerts an anti-stress effect, and this anti-stress effect of CMI-168 may be a contributor to the better performance in learning and memory. Chan and colleagues demonstrated that chicken essence improves short-term memory, which was assessed by form-color associative memory tests in participants with high depression scores [63]. More recently, Suttiwan et al. showed that chicken essence could improve short-term memory among healthy adults experiencing stress in their daily lives [58].

Several possible mechanisms have been suggested for the anti-stress effect of the essence of chicken, including: (1) Increasing glucose utilization rate and levels of insulin in mice subjected to restraint stress; (2) improving plasma lipid metabolism for energy utilization, and; (3) protecting the liver from oxidative damage [61,62]. In the brain, the amygdala is the regulatory center of stress responses. Our unpublished data shows that six weeks of CMI-168 supplementation in young mice is able to improve the performance of amygdala-related fear conditioning (short-term memory of contextual fear conditioning, time fraction of freezing, 74.67 ± 6.52 vs. 56.67 ± 5.07%, CMI-168 vs. Chow, *p* = 0.043). Hence, it is possible that the alteration of amygdalar transmission is involved in the CMI-168 supplement-induced anti-stress effects.

## 5. Conclusions

The effects of CMI-168 dietary supplementation in middle-aged mice on the structure and function of hippocampi at the behavioral and cellular levels were investigated. CMI-168 supplementation led to improvements in learning and memory, as well as resilience to stress in the mice. The hippocampus may be not directly involved in the CMI-168-induced benefits on cognitive function.

## Figures and Tables

**Figure 1 nutrients-11-00027-f001:**
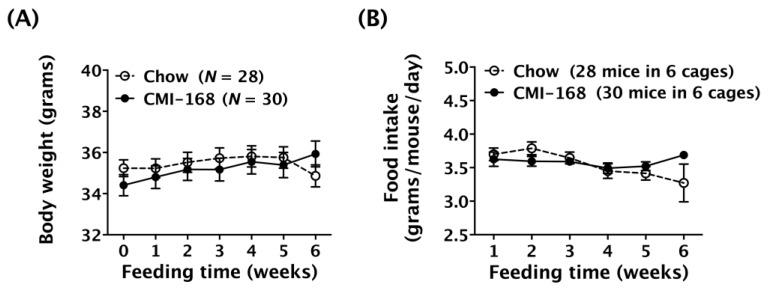
Effects of six-week Chicken Meat Ingredient (CMI-168) supplementation on the gain in body weight and food intake in mice. (**A**) Body weight of mice on the Chow- and CMI-168-supplemented diets over the experimental period. (**B**) Food intake of mice on the Chow- and CMI-168-supplemented diets over the experimental period.

**Figure 2 nutrients-11-00027-f002:**
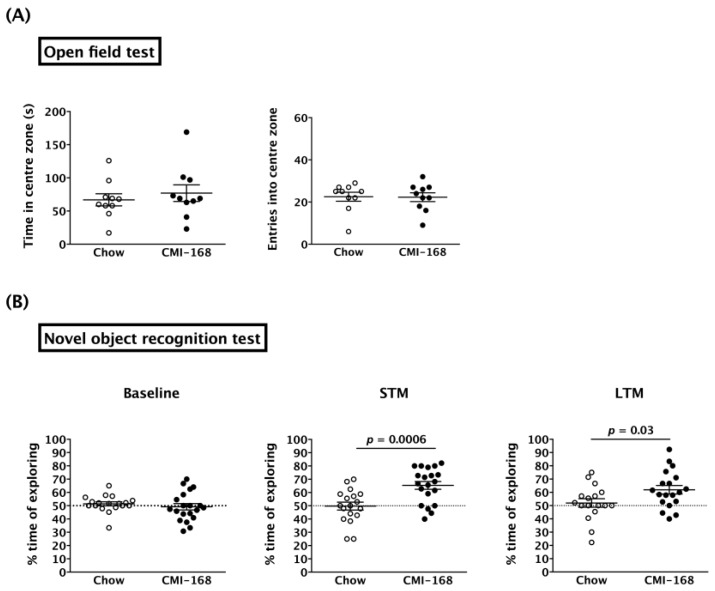
Effects of six-week CMI-168 dietary supplementation on the open field test (OFT) and novel object recognition test (ORT) in mice. (**A**) Time spent in the centre zone and the number of entries into the centre zone for the OFT. (**B**) Percentage of time spent exploring the chosen (in baseline) or novel objects (short-term memory (STM) and long-term memory (LTM)) in the ORT.

**Figure 3 nutrients-11-00027-f003:**
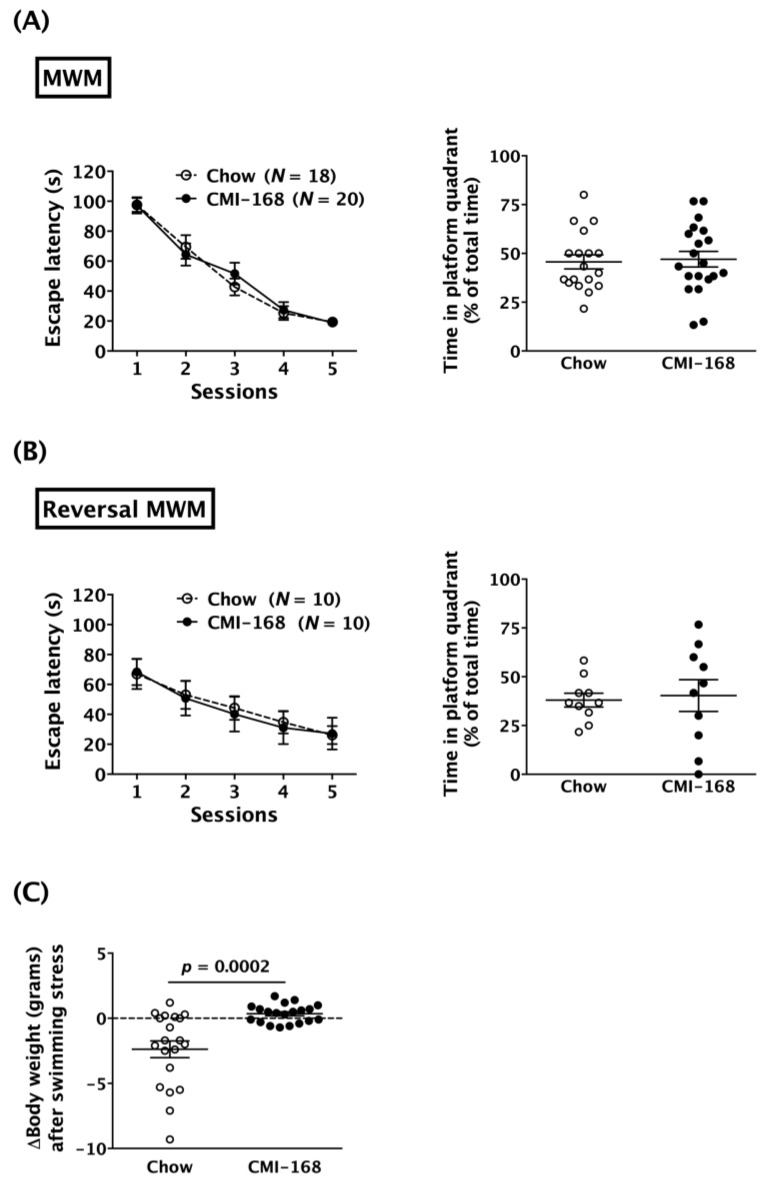
Effects of six-week CMI-168 dietary supplementation on the performances of the Morris water maze (MWM) and swimming stress-induced weight loss in mice. (**A**) Escape latencies during training sessions and the time spent in the platform quadrant in the probe test of the Chow and CMI-168 mice in the MWM. (**B**) Escape latencies during training sessions and the time spent in the platform quadrant in the probe test of the Chow and CMI-168 mice in the reversal MWM. (**C**) Swimming stress-induced body weight changes of the Chow and CMI-168 mice.

**Figure 4 nutrients-11-00027-f004:**
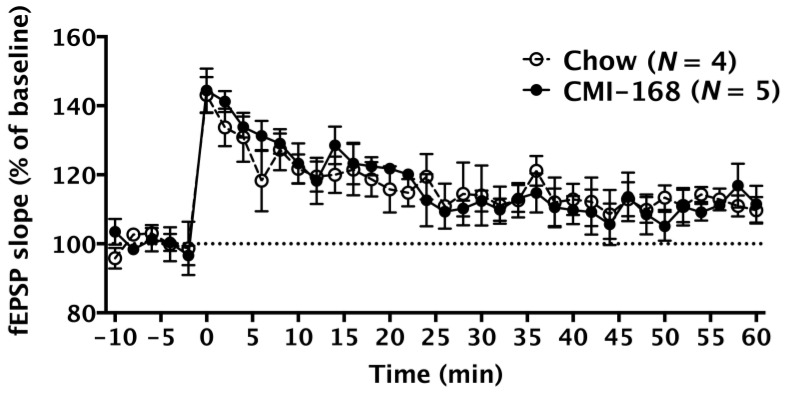
Effects of six-week CMI-168 dietary supplementation on the long-term potentiation (LTP) induction and maintenance in mice. The field excitatory postsynaptic potentials (fEPSP) slope (% of baseline) was recorded in the CA1 region of the Chow and CMI-168 mice before and after the LTP induction. LTP was induced by high-frequency stimulation applied to the hippocampal slices at 0 min on the timeline.

**Figure 5 nutrients-11-00027-f005:**
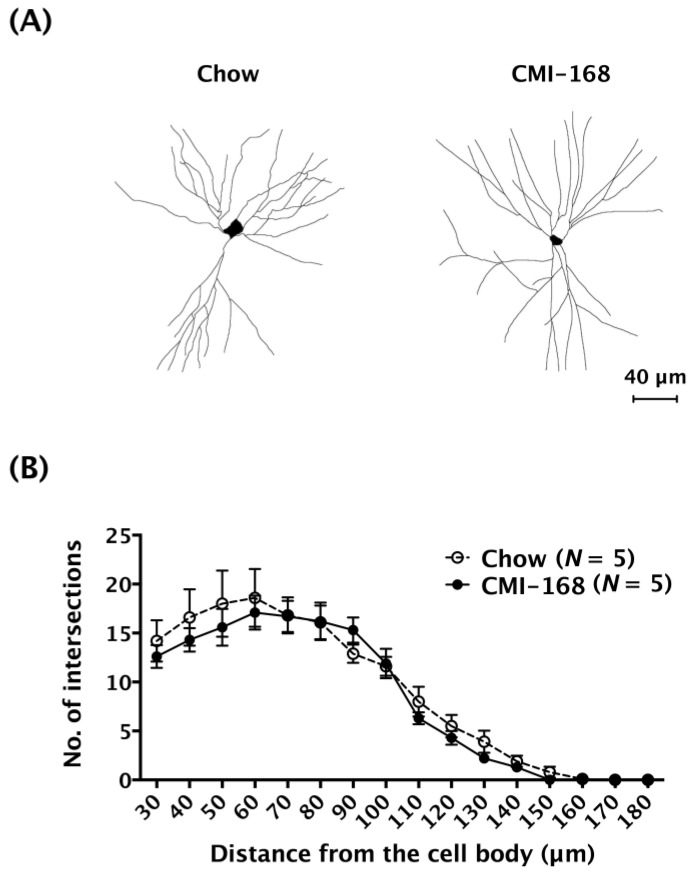
Effects of six-week CMI-168 dietary supplementation on the dendritic complexity of CA1 pyramidal neurons in mice. (**A**) Representative micrographs of the dendritic arborization of CA1 neurons of the Chow and CMI-168 mice. (**B**) Morphometric results of Sholl analysis from the CA1 neurons of the Chow and CMI-168 mice.

**Figure 6 nutrients-11-00027-f006:**
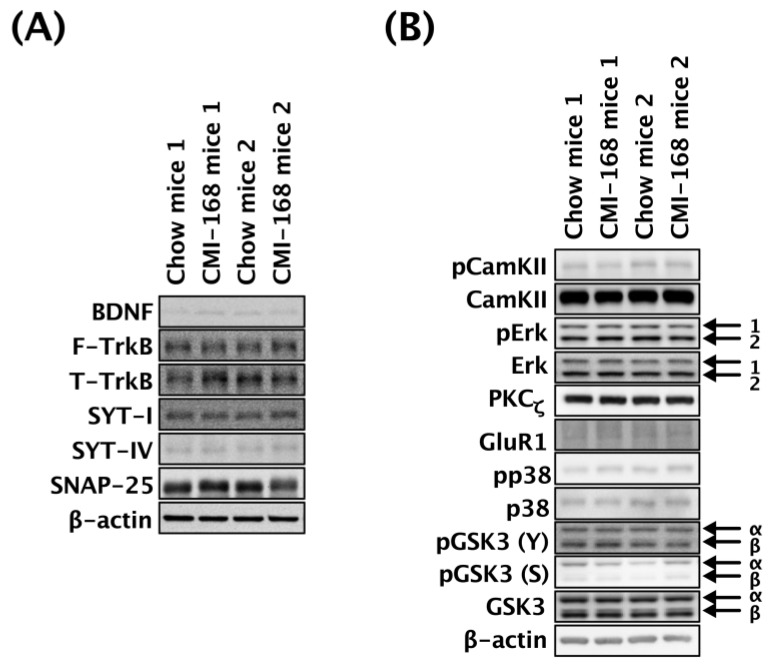
Effects of six-week CMI-168 dietary supplementation on the hippocampal expression or phosphorylation levels of the neuroplasticity-related protein in mice. (**A**) Representative Western blot analysis of hippocampal lysates from Chow- and CMI-168-supplemented mice probed with BDNF, F-TrkB, T-TrkB, SYT-I, SYT-IV and SNAP-25. (**B**) Representative Western blot analysis of hippocampal lysates from Chow- and CMI-168-supplemented mice probed with pCamKII, CamKII, pErk, Erk, PKC_ζ_, GluR1, pp38, p38, pGSK3 (Y), pGSK3 (S) and GSK3.

**Table 1 nutrients-11-00027-t001:** Information of primary antibodies used in this present study.

Antibody	Species/Clonality	Source (Catalogue Number)	Dilution
Anti-BDNF	Rabbit/Polyclonal	Santa Cruz, sc-546	1:500
Anti-TrkB	Mouse/Monoclonal	BD Bioscience, 610101	1:5000
Anti-SYT-I	Mouse/Monoclonal	Stressgen, SYA-130	1:5000
Anti-SYT-IV	Rabbit/Polyclonal	Santa Cruz, sc-30095	1:1000
Anti-SNAP-25	Rabbit/Polyclonal	Stressgen, ADI-VAM-SV012	1:5000
Anti-PKC_ζ_	Goat/Polyclonal	Santa Cruz, sc-216-G	1:1000
Anti-pCamKII	Rabbit/Monoclonal	Cell Signaling Tech, 12716	1:1000
Anti-CamKII	Rabbit/Monoclonal	Cell Signaling Tech, 11945	1:1000
Anti-pErk1/2	Rabbit/ Monoclonal	Cell Signaling Tech, 4370	1:10,000
Anti-Erk1/2	Rabbit/Monoclonal	Cell Signaling Tech, 4695	1:10,000
Anti-GluR1	Rabbit/Monoclonal	Merck-Millipore, 04-855	1:5000
pp-38	Rabbit/Monoclonal	Cell Signaling Tech, 4511	1:1000
p-38	Rabbit/Monoclonal	Cell Signaling Tech, 8690	1:1000
pGSK3α/β (Y)	Mouse/Monoclonal	Merck-Millipore, 05-413	1:1000
pGSK3α/β (S)	Rabbit/Monoclonal	Cell Signaling Tech, 9327	1:1000
GSK3α/β	Rabbit/Monoclonal	Cell Signaling Tech, 5676	1:1000
Anti-β-actin	Mouse/Polyclonal	Sigma-Aldrich, A2228	1:10,000

BDNF: brain-derived neurotrophic factor; TrkB: tyrosine receptor kinase B; SYT: synaptotagmin; SNAP-25: synaptosomal-associated protein-25; PKC: protein kinase C; CamKII: Ca2+/calmodulin-dependent protein kinase II; Erk1/2: extracellular regulated protein kinase 1/2; GluR1: glutamate receptor 1; p38: p38 mitogen-activated protein kinase; GSK3α/β: glycogen synthase kinase 3-α/β.

**Table 2 nutrients-11-00027-t002:** Quantified results (relative expression) of immunoblotting.

Selected Proteins	Chow Mean ± SEM (*n*)	CMI-168 Mean ± SEM (*n*)	*p*-value
BDNF	1.00 ± 0.09 (8)	0.79 ± 0.07 (10)	0.0858
F-TrkB	1.00 ± 0.07 (8)	1.11 ± 0.07 (10)	0.2652
T-TrkB	1.00 ± 0.14 (8)	1.10 ± 0.16 (10)	> 0.5
SYT-I	1.00 ± 0.03 (8)	0.97 ± 0.03 (10)	0.4971
SYT-IV	1.00 ± 0.05 (8)	1.01 ± 0.06 (10)	> 0.5
SNAP-25	1.00 ± 0.14 (8)	0.96 ± 0.12 (10)	> 0.5
pCamKII/CamKII	1.00 ± 0.16 (7)	0.72 ± 0.24 (7)	0.3580
pErk1/Erk1	1.00 ± 0.21 (7)	1.07 ± 0.16 (7)	> 0.5
pErk2/Erk2	1.00 ± 0.21 (7)	1.08 ± 0.19 (7)	> 0.5
PKC_ζ_	1.00 ± 0.20 (7)	1.09 ± 0.30 (7)	> 0.5
GluR1	1.00 ± 0.10 (7)	1.06 ± 0.16 (7)	> 0.5
pp38/p38	1.00 ± 0.24 (7)	0.89 ± 0.23 (7)	> 0.5
pGSK3α (Y)/GSK3α	1.00 ± 0.07 (7)	1.03 ± 0.09 (7)	> 0.5
pGSK3α (S)/GSK3α	1.00 ± 0.16 (7)	0.77 ± 0.11 (7)	0.2752
pGSK3β (Y)/GSK3β	1.00 ± 0.07 (7)	1.03 ± 0.09 (7)	> 0.5
pGSK3β (S)/GSK3β	1.00 ± 0.18 (7)	0.91 ± 0.17 (7)	> 0.5

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
