# Peer review of "A Hydrolyzed Chicken Extract CMI-168 Enhances Learning and Memory in Middle-Aged Mice"

_nutrients, 2018, doi:10.3390/nu11010027_

Round 1

Reviewer 1 Report

This paper investigates the effect of CMI-168 consumption on cognitive processes in mice. The used approach is interesting because observations are conducted at the behavioral, cellular and molecular levels, allowing a general idea of the mechanism. The paper is well presented in general, only lacking some minor clarifications.

The dose of CMI-168 used in mice was converted from the efficacious dose in a previous human study. However it is not clear how the conversion factor is used to go from 679 to 150 mg/kg. It is not clear either why this conversion factor is used, a reference is needed.

The electrophysiological methods need additional information. The composition of the artificial cerebrospinal fluid should be mentioned as well as the type of high-frequency stimulation used (number of pulses and frequency). The condition of incubation of the slices is also not known (interface or submersion).

For the statistical analysis where several time points where used to compare chow and CMI-168 mice (second repeated factor of the two-way ANOVA), the time points used should be indicated.

The authors suggest that CMI-168 protects the mice against swimming-induced weight loss, and the discussion is based on the fact that this weight loss in control mice is due to stress. This link should be more documented as no experimental measures of stress are presented in the result section. The reference 11 is about social instability stress, and not swimming stress.

Typing/spelling error:

l.27 “...no difference...”

l. 43 “...functional changes in this brain region is…”

l. 54 “...at this is the age…”

Author Response

Reviewer 1

Comment 1

The dose of CMI-168 used in mice was converted from the efficacious dose in a previous human study. However, it is not clear how the conversion factor is used to go from 679 to 150 mg/kg. It is not clear either why this conversion factor is used, a reference is needed.

Response

We used this conversion factor according to the “Guidance for Industry-Estimating the Maximum Safe Starting Dose in Initial Clinical Trials for Therapeutics in Adult Healthy Volunteers” released by U.S. Food and Drug Administration. We have cited this guidance (reference 8) in our revised manuscript.

Comment 2

The electrophysiological methods need additional information. The composition of the artificial cerebrospinal fluid should be mentioned as well as the type of high-frequency stimulation used (number of pulses and frequency). The condition of incubation of the slices is also not known (interface or submersion).

Response

Thanks for the suggestions. We have added the essential information into the section of “Materials and Methods-Electrophysiology”:

The mouse brains were quickly removed, placed in chilled artificial cerebrospinal fluid (aCSF, 117 mM NaCl, 4.7 mM KCl, 2.5 mM CaCl2, 11 mM glucose, 1.2 mM MgCl2, 25 mM NaHCO3, 1.2 mM NaH2PO4, pH 7.4) and equilibrated with 95% O2and 5% CO2. Using a vibratome (DTK-1000N, Dosaka, Kyoto, Japan), the brains were cut into 400-μm transverse slices and recovered for 1 h at room temperature in aCSF bubbled with a mixture of 95% O2and 5% CO2. A 64-channel microelectrode array system (MED-P515A, Alpha MED Scientific Inc., Osaka, Japan) was used to record the field excitatory postsynaptic potentials (fEPSP) and introduce the high-frequency stimulation. The brain slices were placed into a submersion chamber equipped with the 64-channel multielectrode probe with the CA1 pyramidal cell layer positioned directly over the array contacts. Recordings were made at 37 °C in circulating aCSF containing 10 μM bicuculline and bubbled with 95% O2and 5% CO2. The fEPSP were captured and analyzed using packed software (Mobius software, Alpha MED Scientific Inc.). There was a 10-min period of pre-induction baseline measures in which stimuli were elicited at 20-s intervals. After the baseline measures, LTP was induced by one train of high-frequency stimulation (100 Hz, 1 s). The fEPSPs were recorded for at least 60 min after the high-frequency stimulation had begun. Stimulation intensity was determined prior to the start of the experiment as the intensity producing a fEPSP amplitude 60% of the maximum response. Each piece of data was obtained from one single hippocampal slice from different mice.

Comment 3

For the statistical analysis where several time points whereused to compare chow and CMI-168 mice (second repeated factor of the two-way ANOVA), the time points used should be indicated.

Response

In this study, repeated measured two-way ANOVAs were used to analyze the data sets of body weights, food intakes, learning curves of MWM, and LTP in mice. Because the aim of this study was to characterize the effect of CMI-168, therefore we only reported the two-way ANOVA results of the CMI-168 factor but not the time factor. Furthermore, the time factor (two-way ANOVA) in the learning curves of MWM and LTP are expected to be significant when the MWM is successfully implemented and LTP is successfully induced. Our results indicated that there were no statistical differences between CMI-168 and Chow groups in these four datasets. Therefore, we did not further perform multiple comparisons to analyze the differences in each time point. 

As requested by the Reviewer, we have included the statistical results of the time factor and post-hoc analyses in the Resultssection in the revised manuscript.

1.In “CMI-168 did not affect body weight or food consumption in middle-aged mice” section:

“…monitored. Results showed that the body weights of the Chow and CMI-168 mice were unchanged during the feeding period (F= 0.95, d.f.6/336, p= 0.46, time factor, repeated two-way ANOVA, Figure 1A). However, the food intakes altered during the feeding period (F= 2.60, d.f.5/50, p= 0.0363, time factor, repeated two-way ANOVA, Figure 1B). The post-hoc test revealed that the food intake of Chow mice in the 6thweek was less than those in the 1st(p= 0.0254) and 2nd(p= 0.0030) week (Figure 1B).”

2.In “CMI-168 did not alter the hippocampus-related spatial memory in middle-aged mice” section:

“…memory. Results showed that the escape latencies of both Chow and CMI-168 mice reduced session by session (F= 79.98, d.f.4/144, p< 0.0001, session factor, repeated two-way ANOVA, Figure 3A). The post-hoc analysis revealed that the escape latencies in the 5thsession were significantly less than those in the 1stsession in both Chow (p< 0.0001) and CMI-168 (p< 0.0001) groups. These data indicated that both CMI-168 and Chow mice successfully learned the location on platform in the MWM test (Figure 3A).” and “…flexibility, both Chow and CMI-168 mice also successfully learned the platform location by showing decreased escape latencies session by session (F= 13.38, d.f.4/72, p< 0.0001, session factor, repeated two-way ANOVA, Figure 3B). Post-hoc analysis revealed that the escape latencies in the 5thsession were significantly less than those in the 1stsession in both Chow (p= 0.0002) and CMI-168 (p= 0.0001) groups in the reversal MWM test (Figure 3B).”

3.In “CMI-168 did not alter the LTP induction, LTP maintenance or the dendritic complexity of the hippocampal CA1 neurons in middle-aged mice” section:

“High-frequency stimulation successfully induced LTP in the middle-aged mice of both the Chow and CMI-168 group (F= 11.16, d.f.35/245, p< 0.0001, time factor, repeated two-way ANOVA, Figure 4).”

Comment 4

The authors suggest that CMI-168 protects the mice against swimming-induced weight loss, and the discussion is based on the fact that this weight loss in control mice is due to stress. This link should be more documented as no experimental measures of stress are presented in the result section. The reference 11 is about social instability stress, and not swimming stress.

Response

We appreciate Reviewer’s comment. We have replaced the reference 11 (social instability stress) by another reference (Ref 57) which reported that repeated swimming stress induced a loss of body weight in mice to support our point. 

Comment 5

Typing/spelling error: l.27 “...no difference...”; l. 43 “...functional changes in this brain region is…”; l. 54 “...at this is the age…”.

Responses

We apologize for these mistakes and have corrected them in our revised manuscript.

Reviewer 2 Report

In this review article “A hydrolyzed chicken extract CMI-168 enhances learning and memory in middle-aged mice”  the authors showed that  the dietary supplement of CMI-168 for 6 weeks, significantly improved  the non-spatial memory in the middle-aged mice. Also, they tried to  understand the molecular mechanism of CMI-168 induced cognitive process.  The experiments were nicely designed, performed  and the manuscript was written well.

Some concerns are following,

The  authors have chosen the dose based on the previous study in human  subjects and performed this study. But, did the authors or any previous  studies showed the effect  of a higher dose? Because, if the authors used different doses, it may  reveal, whether higher dose provides more benefits or side effects?

In Figure 6, the protein SNAP-25 was wrongly denoted as SANP-25. It should be corrected.

In  Figure 6, the authors showed the immunoblotting images in 4 groups (2  Chow and 2 CMI-168). Are these representing the groups in duplicates? If  so, the authors should  mention in the figures.

The authors did not discuss the neuroplasticity related molecules in the discussion section.

Author Response

Reviewer 2

Comment 1

The authors have chosen the dose based on the previous study in human subjects and performed this study. But, did the authors or any previous studies showed the effect of a higher dose? Because, if the authors used different doses, it may reveal, whether higher dose provides more benefits or side effects?

Response

Recently, we have treated the senescence accelerated mouse line, SAMP8 mice, with CMI-168 at dosages of 150 mg/kg (1X), 300 mg/kg (2X) and 600 mg/kg (4X). Results showed that all dosages enhanced learning and memory in passive avoidance and active shuttle response tasks without obvious adverse side effects. We have incorporated these points into the 2ndparagraph of Discussionsection of the revised manuscript.

“In line with this present study, we recently treated senescence accelerated mice with a long-term CMI-168 supplementation at dosages of 150, 300, and 600 mg/kg and found that all dosages enhanced learning and memory in passive avoidance and active shuttle response tasks without obvious adverse side effects [16].”

Comment 2

In Figure 6, the protein SNAP-25 was wrongly denoted as SANP-25. It should be corrected.

Response

We apologize for this mistake and have corrected it in the revised manuscript.

Comment 3

In Figure 6, the authors showed the immunoblotting images in 4 groups (2 Chow and 2 CMI-168). Are these representing the groups in duplicates? If so, the authors should mention in the figures.

Response

Yes, the representing micrographs show the duplicated results of the expression of indicated proteins. We have modified the denotation of Figure 6to clear this point in our revised manuscript.

Comment 4

The authors did not discuss the neuroplasticity related molecules in the discussion section.

Response

We have added some statements to discuss the neuroplasticity related molecules in the 2ndparagraph of Discussionin our revised manuscript: 

“Among the selected neuroplasticity-related proteins, the BDNF-TrkB signaling [17, 18], CamKII [19], Erk1/2 [20], PKCζ [21], GluR1 [22], p38 [23], and GSK3 [24] have been linked to the learning and memory functions. SYT-1, SYT-4, and SNAP-25 are members of SNAREs located in the presynaptic and postsynaptic regions. These synaptic proteins regulate the release of neurotransmitters at the presynaptic terminals and the localization of receptors in the postsynaptic terminals [25]. Reducing the local expression of SYT-1, SYT-4, and SNAP-25 are known to impair learning and memory [26]. Nonetheless, CMI-168 did not affect the expressions of these synaptic proteins or the neuronal structures in the hippocampus. Taken together…”
